# LESS IS MORE: RESOURCE-EFFICIENT LOW-RANK ADAPTATION

## ABSTRACT

Low-Rank Adaptation (LoRA) is a widely adopted parameter-efficient fine-tuning (PEFT) method for Large Language Models (LLMs), but it still incurs notable overhead and suffers from parameter interference in complex datasets. While recent works decouple LoRA update matrices to exploit matrix-wise asymmetry, training costs remain high. We revisit LoRA from the perspective of inter-matrix and intra-layer parameter redundancy and propose Resource-Efficient Low-Rank Adaptation, `ReLoRA`, a lightweight and generalizable approach for language, multimodal, and diffusion models. `ReLoRA` employs a unified A matrix across all transformer layers and introduces a runtime selective B matrices update to dynamically trade-off the system resource budget and model performance. `ReLoRA` consistently outperforms LoRA across diverse modalities, including common-sense reasoning, visual instruction tuning, and image generation, demonstrating improved efficiency and robustness. Anonymous codes are submitted with the paper and will be publicly available.

## 1 INTRODUCTION

Large Language Models (LLMs; Brown et al. 2020; Devlin et al. 2019; AI@Meta 2024; Meta Platforms, Inc. 2024) offer impressive generalization capabilities but are exceedingly costly to train from scratch. Consequently, fine-tuning pretrained LLMs for multiple downstream tasks has emerged as a prevalent technique to meet domain-specific requirements, effectively balancing performance and resource efficiency. However, full fine-tuning (FFT)—which updates every parameter in models consisting of billions of parameters—remains computationally and memory-intensive. To overcome these limitations, Parameter-Efficient Fine-Tuning (PEFT) methods have been proposed, including LoRA (Zhang et al., 2023b; Hu et al., 2022; Liu et al., 2024b), adapters (Rebuffi et al., 2017; Houlsby et al., 2019; Karimi Mahabadi et al., 2021), and various derivatives (Li & Liang, 2021; Lester et al., 2021; Deng et al., 2022; He et al., 2021). PEFT selectively tunes only a subset of model parameters or incorporates specialized modules tailored to specific tasks. By maintaining most of the base model parameters frozen and fine-tuning only a limited number of task-specific parameters, PEFT, like LoRA, substantially decreases computational and memory overhead during both adaptation and deployment phases, thus extending the practical applicability of LLMs. Current research efforts largely aim at enhancing the efficiency of LoRA further, particularly by minimizing the number of trainable parameters (Zhang et al., 2023a; Tian et al., 2024). Nevertheless, excessively aggressive parameter reduction may hinder convergence (Yeh et al., 2023), while overly cautious approaches risk overfitting. Moreover, PEFT methods (Kaplan et al., 2020; Liu et al., 2024b) inherently underperform compared to FFT due to the limited parameter updates, highlighting an essential trade-off between efficiency and performance. This performance gap becomes particularly evident in complex domains characterized by diverse sub-domains and intricate task distributions (Dou et al., 2024; Li et al., 2024). This situation presents a compelling research question:

*How to achieve high performance and efficient fine-tuning across heterogeneous domains within tight resource constraints?*

Recent studies reveal significant parameter redundancy in low-rank adaptation. This redundancy manifests at both the matrix-wise (Zhang et al., 2023a; Song et al., 2024; Kopiczko et al., 2023) and layer-wise (Yao et al., 2024; Lin et al., 2024a; Renduchintala et al., 2023; Pan et al., 2024) levels, as similar adaptation patterns often recur across different modules, leading to inflated parameter

counts. Initial approaches to mitigate this issue involve sharing (Song et al., 2024) or freezing (Zhang et al., 2023a) low-rank matrices across layers. While these techniques reduce parameter overhead, they often do so at the cost of model expressiveness and generality. The tension between efficiency and performance is particularly acute in complex task learning, where one must balance task interference against cross-task synergy. To address this, recent work has explored Mixture-of-Experts (MoE) frameworks (Gao et al., 2024; Tang et al., 2025; Li et al., 2024) or has decoupled adapters into shared and task-specific components (Tian et al., 2024; Hayou et al., 2024). However, such modular designs typically increase the total number of tunable parameters, highlighting an ongoing need for a more efficient trade-off between adaptation capacity and parameter efficiency.

To address these challenges, we introduce Resource-Efficient Low-Rank Adaptation, `ReLoRA`, a lightweight and generalizable framework designed to mitigate both parameter redundancy and interference. In particular, `ReLoRA` tackles redundancy by employing a single, unified low-rank matrix $A$ across all transformer layers. This design enforces a common adaptation subspace, thereby eliminating repetitive per-layer parameters. Meanwhile, `ReLoRA` introduces a selective $B$ matrices update to enhance both efficiency and robustness, further reducing parameter overhead. For complex settings, `ReLoRA` deploys parallel, task-specific "B-heads" that learn distinct transformations while leveraging the shared subspace defined by A. This modular architecture effectively reduces potentially conflicting task objectives, mitigating interference and promoting shared knowledge transfer. The resulting design strikes a principled balance between parameter efficiency and model expressiveness, enabling scalable fine-tuning under stringent resource constraints. Across diverse natural language, multimodal, and diffusion tasks, `ReLoRA` consistently outperforms strong baselines, establishing it as a robust and efficient framework for fine-tuning in complex, heterogeneous scenarios.

## 2 BACKGROUND AND MOTIVATION

### 2.1 LOW-RANK ADAPTATION

Low-Rank Adaptation (LoRA) (Hu et al., 2022) is an efficient fine-tuning technique for large pre-trained models, introducing small low-rank matrices ($A$ and $B$) that can be applied to arbitrary linear layers. Formally, for a linear transformation $h = Wx$ with input $x \in \mathbb{R}^{d_i}$ and weight $W \in \mathbb{R}^{d_o \times d_i}$, LoRA learns a low-rank decomposed update:

$$y' = y + \Delta y = Wx + BAx, \tag{1}$$

where $y \in \mathbb{R}^{d_o}$ is the output, and $A \in \mathbb{R}^{r \times d_i}$, $B \in \mathbb{R}^{d_o \times r}$ are low-rank matrices with $r \ll \min(d_o, d_i)$ as the chosen rank. Typically, $B$ is initialized to zeros, while $A$ follows a Gaussian matrix. During fine-tuning, only $A$ and $B$ are updated, keeping the original model parameters frozen, thus significantly reducing computational overhead.

### 2.2 OBSERVATIONS

In this subsection, we revisit LoRA to analyze the trade-off between expressiveness and parameter efficiency and conduct systematic experiments that shed light on its underlying mechanisms.

**Observation I:** *LoRA exhibits significant parameter redundancy at both the inter-matrix and intra-layer level.* ***Inter-matrix***: for a single LoRA adapter with matrices *A* and *B*, recent studies (Tian et al., 2024; Hayou et al., 2024) observe that that the down-projection matrix $A$ converges to a strikingly similar subspace across different layers. Consequently, strategies like freezing (Zhang et al., 2023a) and sharing (Song et al., 2024) the $A$ matrix after initialization can effectively capture this common basis while eliminating redundant parameters. As shown in Figure 1, both approaches perform comparably to—and sometimes slightly better than—vanilla LoRA, confirming the high degree of inter-matrix redundancy. ***Intra-layer***: Prior work has established that different layers in an LLM contribute unequally to fine-tuning, with adaptation often concentrated in a small subset of layers (Yao et al., 2024; Lin et al., 2024a; Renduchintala et al., 2023; Pan et al., 2024). Building on this insight, we find that the LoRA matrices themselves exhibit a similar pattern of varying importance. To demonstrate this, we adopt a shared-$A$ design and randomly prune $N$ layer-specific $B$ matrices. For a 32-layer model, this reduces the parameter budget from $(A + B) \times 32$ to $A + B \times (32 - N)$. As illustrated in Figure 2, experiments on Llama-3-8B show that discarding 50%

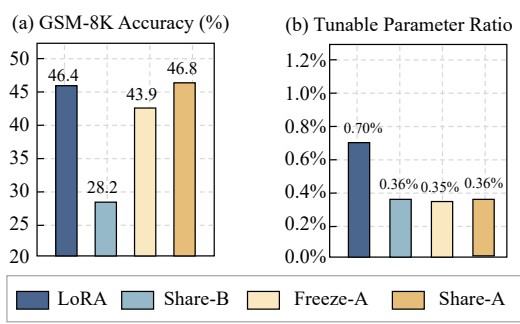 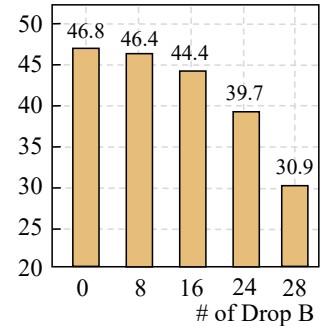

Figure 1: Matrix-wise optimization of LoRA.

Figure 2: Impact of dropping different numbers of B modules.

of the B matrices degrades performance by a mere 2.4%. This result indicates a long-tailed utility distribution, where a large fraction of layer-specific adapters are expendable and can be pruned with minimal impact on performance.

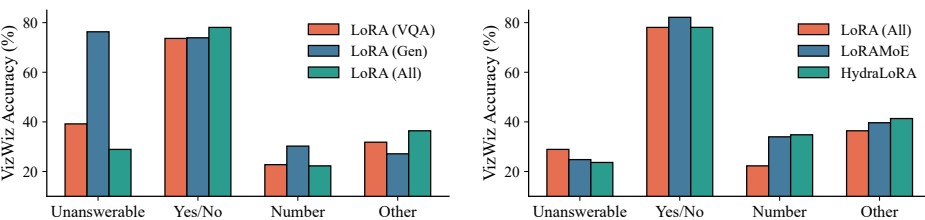

Figure 3: Performance comparison on heterogeneous data on LLaVA-7B (Liu et al., 2023a), evaluated on the VizWiz dataset (Bigham et al., 2010).

**Observation II:** *Fine-tuning on heterogeneous data reveals a tension between task-specific conflicts and latent cross-domain commonalities.* We illustrate this tension by fine-tuning LLaVA-v1.5-7B (Liu et al., 2023a) on a mixed dataset from two distinct domains: Visual Question Answering (VQA) (Antol et al., 2015) and open-ended Generation (Gen) (Liu et al., 2023a; Mostafazadeh et al., 2016). As detailed in Figure 3, evaluating on the VizWiz benchmark (Bigham et al., 2010) reveals that naively combining these domains forces a single set of adapter parameters to learn conflicting objectives, leading to significant performance degradation and optimization interference. This suggests that domain signals must be modulated carefully. A Mixture-of-Experts (MoE) approach, which routes inputs to specialized LoRA adapters, can mitigate task conflicts and even exceed single-domain performance on some metrics (e.g., 82.15% on Yes/No questions). However, this hard partitioning can fail to leverage shared knowledge, causing it to underperform on others (e.g., Unanswerable). One more balanced strategy like HydraLoRA (Tian et al., 2024), which shares a global down-projection matrix A while maintaining task-specific up-projection matrices B, better captures both commonalities and specializations. This architecture achieves the highest mean score (38.10%), surpassing both MoE-LoRA (37.44%) and vanilla LoRA (36.00%). Nevertheless, its reliance on separate per-task B matrices substantially increases the parameter budget. This leaves open the challenge of achieving cross-task synergy without the high overhead of explicit expert modules.

## 3 ReLoRA

Resource-Efficient Low-Rank Adaptation (ReLoRA) is designed to address the parameter redundancy and high training costs inherent in standard LoRA. As illustrated in Figure 4, ReLoRA achieves this through two core innovations: (1) a Unified Asymmetric Architecture that maximizes parameter efficiency through cross-layer sharing, and (2) a dynamic training Reducer that intelligently and selectively updates parameters during training to balance model performance with computational resources.

Figure 4: Architecture and workflow of ReLoRA. Given a base model and a target dataset, the *Configurator* generates a shared-asymmetric-head LoRA structure, where a global low-rank matrix $A$ is reused across layers while each $B_{i,j}$ remains layer- and head-specific. A *Reducer* then prunes redundant $B$ heads under resource and performance constraints, yielding an optimized low-parameter LoRA configuration that balances efficiency and effectiveness.

## 3.1 UNIFIED ASYMMETRIC ARCHITECTURE FOR PARAMETER EFFICIENCY

The foundation of ReLoRA is its novel parameter-sharing structure, designed to tackle both intra-layer and inter-matrix redundancy. This architecture is established by a Configurator responsible for initializing the model's low-rank matrices.

**Global Knowledge Sharing via a Unified Matrix A.** The Configurator first initializes a single, globally shared low-rank matrix $A \in \mathbb{R}^{d \times r}$ that is reused across all Transformer layers. Unlike traditional LoRA, which allocates unique A and B matrices for each layer, our approach drastically reduces the total number of trainable parameters. This shared matrix A is designed to capture and encode generalizable, model-wide knowledge, forming a highly parameter-efficient backbone for adaptation.

**Input-Specific Adaptation with a Dynamic Router and Expert Matrices B.** Complementing the shared matrix A, the Configurator initializes a set of multiple low-rank "expert" matrices $\{B_i^{(n)}\}_{i=1}^m$ for each Transformer layer n. These expert matrices, $B_i^n \in \mathbb{R}^{d \times r}$, are designed to capture fine-grained, specialized knowledge specific to each layer.

To enable dynamic, input-aware adaptation, we introduce a lightweight Router network. The Router's role is to dynamically select which experts to activate for each input token during both training and inference. Its architecture includes a dense layer with a trainable weight matrix $W_g \in \mathbb{R}^{r \times N}$. For an intermediate input token representation $x$, the router performs a linear transformation $z = W_g^T x$ and applies a softmax function to convert the output $z$ into normalized gating scores $w_i(x)$. These scores modulate the contribution of each expert. The weight update $\Delta W^n$ for layer $n$ is thus defined as:

$$\Delta W^{(n)} = \left( \sum_{i=1}^m w_i^{(n)} B_i^{(n)} \right) \cdot A, \tag{2}$$

The final adapted weight is given by $W'^{(n)} = W^{(n)} + \Delta W^{(n)}$. This asymmetric design, a static, shared A and a dynamic mixture of expert B matrices, enables both expressive adaptation and parameter efficiency.

## 3.2 REDUCER FOR RESOURCE-AWARE TRAINING

On top of this efficient architecture, we introduce the Reducer, a dynamic training mechanism that minimizes computational overhead by freezing specific B matrices during training. Crucially, this is not a post-training pruning method but a dynamic freezing strategy applied during the training process itself. This online adaptation adaptively reduces the number of active parameters, differing fundamentally from static pruning techniques. The Reducer's core is an importance score vector s, which is updated iteratively to reflect each layer's contribution to the task objective. The scores are calculated through the following process: 1) *Layer Suppression*: In each update step, we select a fixed number of layers (e.g., n-layers-suppressed=16) with the lowest current importance scores.

These layers are temporarily "suppressed" by scaling their outputs to near-zero. 2) *Loss Evaluation*: We then compute the loss on a mini-batch of validation data. A larger increase in loss relative to the baseline (without suppression) indicates that the suppressed layers are more important, as their temporary removal significantly harms performance. 3) *Score Update*: The importance scores of the suppressed layers are updated proportionally to the magnitude of the observed loss increase. This process repeats periodically, refining the scores to reflect each layer's evolving contribution.

At each training step, these importance scores guide the parameter updates. The vector is passed through a Sigmoid function to create a sampling distribution:

$$\mathbf{p} = \sigma(-\mathbf{s})$$

which biases selection towards higher-importance layers. A subset of $K$ layers is then sampled, and only the $B_i^{(n)}$ matrices within these selected layers are updated:

$$B_i^{(n)} \leftarrow \begin{cases} B_i^{(n)} - \eta \nabla_{B_i^{(n)}} \mathcal{L}, & \text{if layer } n \text{ is sampled} \\ \text{frozen}, & \text{otherwise} \end{cases} \tag{3}$$

The hyperparameter $K$ is a critical control knob that allows users to flexibly trade off performance against computational resources. The impact of $K$ is systematically investigated in the experiments (see Figure 5) that demonstrate the robustness.

## 4 EXPERIMENTS AND ANALYSIS

In this section, we detail the principal experiments. To evaluate the effectiveness and robustness of `ReLoRA`, we test it in different modalities—commonsense reasoning (Section 4.1), visual instruction tuning (Section 4.2), and image generation (Section 4.3). We then summarize the key results and provide a concise interpretation.

### 4.1 COMMONSENSE REASONING

#### 4.1.1 EXPERIMENT SETTING

**Model and Dataset.** We fine-tune the LLaMA3-8B model (AI@Meta, 2024) for commonsense reasoning tasks. We first fine-tune the model on Commonsense-170k samples from Hu et al. (2023), and subsequently evaluated on eight widely used benchmarks: ARC (Clark et al., 2018), OBQA (Mihaylov et al., 2018), PIQA (Bisk et al., 2019), SIQA (Sap et al., 2019), BoolQ (Clark et al., 2019), HellaSwag (Zellers et al., 2019), and Winog. (Sakaguchi et al., 2021). A detailed description of the dataset can be found in Appendix A.1.

**Baselines.** First, we compare `ReLoRA` with ***different LoRA variants***, including 1) *LoKr* (Yeh et al., 2023) which employs Kronecker products for matrix decomposition of $\mathbf{AB}$; 2) *NoRA* (Lin et al., 2024a) which introduces a dual-layer nested structure with SVD-based initialization, freezing outer LoRA weights, and training an inner LoRA layer. 3) *AdaLoRA* (Zhang et al., 2023b) which parameterizes the incremental updates of the pre-trained weight matrices in the form of singular value decomposition; Second, we extend the experiments exploring `ReLoRA` with ***multi-LoRA optimization*** approaches, including: 4) *HydraLoRA* (Tian et al., 2024): Introduces an asymmetric LoRA architecture with a shared matrix A and multiple distinct B matrices, combined through a trainable MoE router to dynamically adapt to different tasks without requiring domain expertise. 5) *MoLA* (Gao et al., 2024): A parameter-efficient tuning method that integrates LoRA and Mixture-of-Experts (MoE) with layer-wise expert allocation. 6) *LoRAMoE* (Dou et al., 2024): A parameter-efficient fine-tuning method combining LoRA and MoE, freezing the backbone model and introducing experts. 7) *MixLoRA* (Li et al., 2024): A resource-efficient parameter tuning method combining LoRA and MoE with independent attention-layer adapters and load balancing, enhancing multi-task performance and reducing computation and memory costs. 8) *GraphMoE* (Tang et al., 2025): A novel MoE-based architecture that enhances language model reasoning through a self-rethinking mechanism and recurrent routing on a pseudo graph of expert nodes. A detailed description of the baselines and hyperparameter settings can be found in Appendix B.1.

Table 1: Comparative performance of various methods fine-tuning LLaMA3-8B on the commonsense reasoning tasks. * denotes results from the original paper; [1] from (Wu et al., 2024); [2] from(Tang et al., 2025).

| Schemes | ARC-e | OBQA | SIQA | ARC-c | WinoG. | PIQA | BoolQ | HellaS | Avg. | Param. |
|---|---|---|---|---|---|---|---|---|---|---|
| LoRA (Hu et al., 2022) | 84.2 | 79.0 | 79.9 | 71.2 | 84.3 | 85.2 | 70.8 | 91.7 | 80.8 | 0.35% |
| LoRA-FA (Zhang et al., 2023a) | 86.1 | 81.0 | 79.5 | 73.4 | 83.8 | 84.2 | 69.0 | 93.4 | 81.3 | 0.17% |
| ShareLoRA (Song et al., 2024) | 87.5 | 83.1 | 80.2 | 75.0 | 84.0 | 85.5 | 71.0 | 96.1 | 82.8 | 0.18% |
| LoKr[1] (Yeh et al., 2023) | 89.2 | 81.8 | 78.7 | 76.7 | 82.1 | 81.6 | 65.1 | 92.0 | 80.9 | 0.01% |
| NoRA* (Lin et al., 2024a) | 88.2 | 85.0 | 79.1 | 77.5 | 84.3 | 86.4 | 73.3 | 94.1 | 83.1 | 0.09% |
| AdaLoRA[1] (Zhang et al., 2023b) | 90.4 | 85.0 | 76.7 | 79.1 | 83.3 | 86.4 | 75.1 | 75.4 | 81.4 | 0.35% |
| ReLoRA (Single B) | 89.8 | 86.6 | 80.5 | 79.9 | 84.4 | 88.3 | 72.7 | 94.7 | 84.6 | 0.18% |
| HydraLoRA (Tian et al., 2024) | 92.4 | 87.0 | 82.6 | 81.9 | 87.8 | 88.0 | 73.6 | 96.2 | 86.1 | 0.93% |
| MoLA[2] (Gao et al., 2024) | 86.4 | 84.4 | 76.4 | 77.9 | 83.3 | 86.7 | 74.0 | 93.9 | 82.9 | 2.70% |
| LoRAMoE[2] (Dou et al., 2024) | 87.8 | 85.0 | 74.8 | 79.5 | 83.4 | 87.1 | 72.4 | 94.8 | 83.5 | 3.20% |
| MixLoRA* (Li et al., 2024) | 86.5 | 84.8 | 78.8 | 79.9 | 82.1 | 87.6 | 75.0 | 93.3 | 83.5 | 3.00% |
| GraphMoE* (Tang et al., 2025) | 90.3 | 88.2 | 79.4 | 80.6 | 83.7 | 88.8 | 75.9 | 95.3 | 85.3 | 5.90% |
| ReLoRA (Multiple B) | 92.9 | 87.0 | 81.7 | 81.8 | 88.4 | 89.6 | 74.1 | 95.8 | 86.4 | 0.53% |

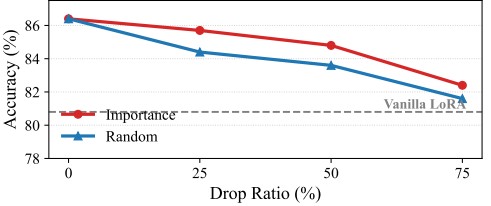

Figure 5: Performance of different drop ratios.

| Method | Average Acc. (%) |
|---|---|
| 1B | 81.3 |
| 2B | 82.7 |
| 3B | 85.1 |
| 4B | 86.4 |

Figure 6: Effect of different numbers of B matrices on model performance

### 4.1.2 PERFORMANCE ANALYSIS.

As shown in Table 1, ReLoRA with multiple B achieves an average accuracy of 86.4% on commonsense reasoning tasks while updating only 0.53% of the backbone parameters. This result surpasses other strong methods like HydraLoRA (86.1%, 0.93%) and GraphMoE (85.3%, 5.90%). Notably, compared to HydraLoRA, ReLoRA achieves a 0.3% higher accuracy with nearly 43% fewer tunable parameters (0.53% vs. 0.93%). Even a more parameter-efficient version, ReLoRA (Single B), maintains a competitive average accuracy of 84.6% using only 0.18% of the parameters. These results highlight the strong parameter efficiency of ReLoRA, validating that significant redundancy exists across layers and can be exploited without sacrificing performance. Specifically, the shared low-rank matrix A across all layers captures global, task-agnostic representations, while the layer-specific $B_i$ heads concentrate expressive capacity where needed. Additionally, the probabilistic layer sampling mechanism ensures that updates are dynamically allocated to the most contributive layers, counteracting the adverse effects of aggressive parameter reduction. Notably, ReLoRA achieves this without altering vanilla LoRA's internal architecture. This synergy between architectural asymmetry and adaptive update allocation enables ReLoRA to extend the existing LoRA variants of PEFT, offering a principled balance between expressiveness and compression.

### 4.1.3 FRAMEWORK ANALYSIS

**Impact of different drop ratios.** Figure 5 illustrates the performance degradation under varying drop ratios of the trainable LoRA B matrices. For this, we compare two strategies: random dropping and a proposed importance-based dropping. The results clearly show that the importance-based method consistently outperforms random dropping across all sparsity levels. Notably, with importance-based pruning, ReLoRA maintains stable accuracy even as up to 75% of the B matrices are removed. This robustness confirms the long-tailed utility distribution of layer-wise LoRA updates—only a small subset of layers contribute disproportionately to downstream performance. The observed resilience stems from two design principles of ReLoRA: first, the shared global matrix A effectively preserves core semantic representations even as layer-specific parameters are pruned; second, the selective update mechanism adaptively concentrates updates on high-importance layers, mitigating the adverse effects of aggressive parameter reduction.

Table 2: Overhead analysis of fine-tuning with different LoRA approaches.

| Method | Param. | Train time | Relative FLOPs | Performance |
|---|---|---|---|---|
| LoRA (rank=16) | 28.3M | 8.0h | 1.00 | 80.8 |
| LoRA (rank=32) | 56.6M | 14.6h | 3.63 | 83.3 |
| LoRA (rank=64) | 113.2M | 30.4h | 15.11 | 82.7 |
| ReLoRA (rank=16 × 4) | 42.8M | 12.8h | 2.86 | 85.7 |

**Impact of B matrices number.** Table 1 shows the results of an ablation study on the number of task-specific B matrices used from the start. The data reveals a clear positive correlation between the number of B-heads and model performance. The model's average accuracy steadily increases from 81.3% with a single B-head (1B) to 82.7% (2B), 85.1% (3B), and finally 86.4% with four B-heads (4B). While performance consistently rises, the incremental gains suggest diminishing returns as more heads are added. This trend suggests that while adding B-heads increases the model's expressive capacity, the most impactful task knowledge is acquired by the first few heads, with later additions contributing more marginally.

**Impact of Configurator.** As shown in Table 1, disabling the Configurator and using only importance-based selective updates, ReLoRA (single B), yields lower performance (84.6%) despite training 0.18% of the model. This highlights a key limitation of purely importance-driven sparsity: it lacks architectural asymmetry and fails to capture shared structure across tasks. In contrast, ReLoRA's asymmetric design, with a globally shared $A$ matrix and specialized $B_i$ heads, explicitly disentangles generalizable semantics from task-specific variation, enabling joint learning of cross-task commonality and local specialization.

**Impact of Reducer.** We evaluated the benefit of structured parameter dropping via our Reducer component. In contrast to HydraLoRA, which utilizes a full set of 32 down-projection A matrices, ReLoRA employs a single shared A matrix and further prunes the multi-head B matrices using an importance-based strategy. This combined reduction method lowers the tunable parameter count from 0.70% to 0.53% while yielding a 0.3% increase in accuracy. Further analysis, presented in Figure 5, validates our approach by showing that importance-guided dropping consistently outperforms random dropping at all sparsity levels. This confirms the effectiveness of our scoring strategy in preserving the most contributive parameters during compression.

**Overhead Analysis** As shown in Table 2, ReLoRA demonstrates superior efficiency, outperforming the equivalent-rank LoRA (rank=64) with a performance score of 85.7 (vs. 82.7), despite requiring only 42.8M parameters (just 38% of LoRA-64) and 12.8h of training time (a 58% reduction). Furthermore, compared to the parameter-similar LoRA (rank=32), it reduces relative FLOPs (Woo et al., 2025) by 21.2% (2.86 vs. 3.63) while also achieving higher performance. This establishes ReLoRA as a highly efficient and scalable fine-tuning solution that strikes a superior performance-to-cost trade-off.

Table 3: Comparative performance of various methods fine-tuning LLaVA-v1.5-7B.

| Methods | Dataset | MMBench | MMVet | MME | AI2D | DocVQA | MathVista | Avg |
|---|---|---|---|---|---|---|---|---|
| | General | 55.90 | 35.00 | 66.38 | - | - | - | - |
| LoRA | Doc | - | - | - | 50.26 | 31.59 | - | - |
| | Math | - | - | - | - | - | 16.80 | - |
| LoRA | All | 51.40 | 32.90 | 48.52 | 48.83 | 30.71 | 17.70 | 38.34 |
| MoLE | All | **59.70** | 31.90 | 66.05 | 52.78 | 31.42 | 18.30 | 43.35 |
| HydraLoRA | All | 56.70 | **35.70** | 62.89 | 52.78 | 31.67 | 19.10 | 43.14 |
| ReLoRA | All | 58.10 | 34.40 | **68.01** | **52.82** | **32.08** | **19.60** | **44.18** |

## 4.2 VISUAL INSTRUCTION TUNING

**Experiment Setting.** *Model and Dataset.* To evaluate performance on multimodal tasks, we fine-tune the LLaVA1.5-7B (Liu et al., 2023a) using the subset of LLaVA-OneVision single-image (Liu et al., 2024a) dataset, which includes general, document, and math tasks. Square sampling is applied

Figure 7: Comparison of text-to-image generation results. ReLoRA demonstrates superior prompt fidelity over the base model, LoRA, and HydraLoRA

Table 4: Comparative performance of fine-tuning Diffusion.

| Scheme | Quality | Detail | Theme | Creativity | Style | Emotion | Tech | Avg. |
|--------|---------|--------|-------|------------|-------|---------|------|------|
| LoRA | 8.24 | 6.97 | 8.72 | 6.95 | 8.07 | 6.84 | 7.66 | 7.63 |
| HydraLoRA | 8.22 | 6.93 | 8.81 | 7.12 | 8.10 | 6.95 | 7.66 | 7.68 |
| ReLoRA | **8.32** | **7.03** | **8.86** | **7.14** | **8.16** | **7.00** | **7.81** | **7.76** |

Table 5: HPS v2 Scores.

| Method | Avg. HPS v2 (↑) |
|--------|-----------------|
| LoRA | 24.08 |
| HydraLoRA | 24.29 |
| ReLoRA | **24.80** |

to ensure balanced coverage across subsets, promoting better generalization and task diversity. After fine-tuning, we evaluate the model on several benchmarks spanning three categories: general-related (MMBench (Liu et al., 2023c), MMVet (Yu et al., 2024), MME (Fu et al., 2024)), document-related (AI2D (Kembhavi et al., 2016), DocVQA (Mathew et al., 2021)), and math-related (MathVista (Lu et al., 2024)). To mitigate varying score ranges, we are bringing all datasets to a 0–100 scale. A detailed description of the dataset can be found in Appendix A.2.

*Baselines.* We compare ReLoRA against the following baselines: 1) *Single LoRA*, which fine-tunes a single LoRA on each individual dataset. 2) *Multi-LoRA* fine-tunes a LoRA on the combined mixture dataset. 3) *LLaVA-MoLE* (Chen et al., 2024), which integrates lightweight LoRA experts via the MoE framework across different datasets and configurations. 4) *HydraLoRA* (Tian et al., 2024), trained on the combined mixture dataset.

**Performance Analysis.** As shown in Table 3, vanilla LoRA fine-tuned on *single* data sources yields reasonable performance—achieving 50.26 on AI2D and 31.59 on DocVQA when trained solely on document data, and 16.80 on MathVista when trained only on math data. However, joint training across all modalities causes a pronounced performance collapse, with the average score dropping to 38.34, reflecting substantial cross-task interference. In contrast, ReLoRA boosts the joint average to 44.18 and achieves consistent gains across most tasks (e.g., 58.10 on MMBench, 68.01 on MME, and 52.82 on AI2D), demonstrating superior conflict mitigation. These improvements arise from the asymmetric adapter design: a globally shared low-rank matrix A captures universal semantic patterns, while input-conditioned $B_i$ matrices provide localized task-specific capacity. This decoupling suppresses redundancy and enables ReLoRA to maintain both generality and adaptability, ensuring robust performance in multi-task fine-tuning scenarios.

### 4.3 DIFFUSION GENERATION

**Experiment Setting.** *Model and Dataset.* To evaluate performance on image generation tasks, we adopt Stable Diffusion v1.5 (Rombach et al., 2022) as the base model. We fine-tuned the model on the `pokemon-blip-captions` dataset (ModelScope, 2024). For evaluation, we sample 100 prompts to generate images and assess their quality using GPT-as-judge. A detailed description of the dataset and evaluation protocol are provided in Appendix A.3 and Appendix B.3, respectively.

**Performance Analysis.** As demonstrated in Table 4, ReLoRA achieves a top-tier average score of 7.76 when fine-tuning Diffusion v1.5, decisively outperforming both LoRA and HydraLoRA baselines. It delivers consistent improvements across key dimensions, including image quality (8.32),

theme relevance (8.86), and creativity (7.14). This quantitative superiority is further corroborated by the HPS v2 benchmark (Wu et al., 2023) (Table 5), where `ReLoRA` again attains the highest score (24.80), confirming its state-of-the-art image generation quality. These strong empirical results are highlighted in qualitative comparisons (Figure 7), where `ReLoRA` more faithfully renders nuanced details—such as a cartoon butterfly with a sad expression—compared to competing methods. These comprehensive performance gains stem directly from `ReLoRA`'s architectural innovations. The asymmetric adapter design leverages a globally shared low-rank matrix A to encode common generative structures, while dynamically combining input-specific $B_i$ matrices to inject instance-level variability. This separation of concerns allows the model to generalize effectively while retaining expressiveness, minimizing redundancy and enhancing efficiency without sacrificing output quality.

## 5 RELATED WORK

**LoRA and its Variants.** Low-Rank Adaptation (LoRA) (Hu et al., 2022) reduces fine-tuning costs by injecting trainable low-rank matrices into pre-trained weights. Follow-up work (Hayou et al., 2024; Lin et al., 2024b; Valipour et al., 2023; Zhang et al., 2023b; Liu et al., 2024b; Yao et al., 2024) improves either optimization or compression: *(1) Training-centric variants* improve optimization via adaptive learning rates (Hayou et al., 2024) or stochastic regularization (Lin et al., 2024b). *(2) Capacity-centric variants* adjust rank on the fly—e.g., DyLoRA and AdaLoRA dynamically allocate dimensions to balance expressiveness and compactnessValipour et al. (2023); Zhang et al. (2023b). *(3) Structure-centric variants* redesign the decomposition itself: Kronecker (LoKr) and Hadamard (LoHa) (Yeh et al., 2023) factorizations, Tucker cores (FLoRA) (Si et al., 2024), and magnitude–direction splits (DoRA)(Liu et al., 2024b) yield tighter compressions. Complementary efforts share or freeze matrices to curb redundancy—ShareLoRA (Song et al., 2024) flexibly ties $A$ and $B$ across layers, whereas LoRA-FA (Yao et al., 2024) freezes $W$ and $A$, updating only $B$ for minimal memory use. Collectively, these efforts underscore a shift toward highly compact, modular PEFT frameworks that balance expressiveness with stringent resource constraints.

**Multi-LoRA Architecture.** Building on LoRA's success, recent work has moved from a *single* adapter to *collections* of LoRAs that can be composed or routed on demand, aiming to retain low-rank efficiency while boosting flexibility. Early efforts such as LoraHub(Huang et al., 2023) pre-train a pool of domain-specialized adapters and select the best subset at inference time, whereas Multi-LoRA(Wang et al., 2023) "horizontally" slices each LoRA along the rank dimension and equips the slices with learnable scaling factors, increasing expressiveness without inflating parameters. To curb the memory surge that accompanies broad instruction tuning, the Mixture-of-LoRA framework (Zadouri et al., 2023) mixes lightweight adapters to achieve a better accuracy–efficiency trade-off. Subsequent work incorporates explicit expert routing: LoRAMoE (Dou et al., 2024) and MOELoRA(Liu et al., 2023b) place LoRA experts in a Mixture-of-Experts scaffold to shield pre-trained knowledge from conflicting instructions, with the latter targeting medical NLP tasks. From a deployment standpoint, S-LoRA (Sheng et al., 2023) proposes a serving framework that caches and composes multiple adapters with minimal overhead. Most recently, HydraLoRA(Tian et al., 2024) removes the need for manual domain assignment by introducing an *asymmetric* design—a single shared down-projection matrix and per-expert up-projections—thereby pushing parameter efficiency beyond symmetric multi-LoRA baselines. Collectively, these advances demonstrate that carefully orchestrated ensembles of low-rank adapters can deliver scalable, conflict-aware, and resource-friendly adaptation for large language models across diverse scenarios.

## 6 CONCLUSION

In this paper, we introduced `ReLoRA`, a resource-efficient low-rank adaptation framework designed to reduce parameter overhead and mitigate task interference. By revisiting LoRA from the perspective of parameter redundancy, `ReLoRA` employs a unified cross-layer A matrix complemented by a dynamic, selective update mechanism for the B matrices. This architecture not only achieves substantial parameter savings but also enhances model performance and robustness. Extensive experiments across diverse modalities—including language, vision-language, and diffusion models—demonstrate that `ReLoRA` consistently outperforms standard LoRA in both task accuracy and efficiency. More discussion about limitations is available in Appendix C.

## ETHICS STATEMENT

We affirm adherence to the ICLR Code of Ethics. This work studies resource-efficient low-rank adaptation and does not involve human subjects, personally identifiable information, or sensitive attributes. All datasets and pretrained weights used are publicly available and were accessed and used in accordance with their licenses and terms of use; no data scraping outside the providers' terms was performed. We disclose our use of LLM-based writing assistance in a separate LLM-usage section in Appendix D. Potential risks include lowering the computational barrier for deploying more capable models in resource-constrained settings; to mitigate misuse concerns, we evaluate only on standard public benchmarks, refrain from releasing domain-specific models for sensitive applications, and provide documentation to support responsible use. The authors take full responsibility for the integrity and accuracy of the reported results.

## REPRODUCIBILITY STATEMENT

We place strong emphasis on the transparency and reproducibility of our work. To facilitate independent verification, the complete implementation has been provided in the supplementary materials, allowing readers to directly reproduce the reported experiments. In addition, Section 4 of the main text outlines the experimental pipeline, including dataset preparation, model configurations, and training procedures. For further clarity, Appendix B documents the full set of hyperparameter choices and auxiliary details. Together, these resources ensure that our results can be reliably replicated and extended in future research.

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

# A  DATASETS

## A.1  COMMONSENE REASONING

Table 6 presents detailed information about the datasets used in our experiments, including their task names, respective domains, the number of training and test sets, and task types. The details of the benchmarks are as follows:

- BoolQ (Clark et al., 2019): yes/no questions which are naturally occurring and generated in unprompted and unconstrained settings. There are 3270 questions in the test set.
- PIQA (Bisk et al., 2019): questions with two solutions requiring physical commonsense. There are 1830 questions in the test set.
- HellaSwag (Zellers et al., 2019): commonsense NLI questions including a context and several endings which complete the context. There are 10042 questions in the test set.
- WinoGrande (Sakaguchi et al., 2021): fill-in-a-blank task with binary options to choose the right option for a given sentence, which requires commonsense reasoning. There are 1267 questions in the test set.
- ARC-easy (Clark et al., 2018) & ARC-challenge (Clark et al., 2018): the Challenge Set and Easy Set of ARC dataset of genuine grade-school level, containing 2376/1172 multiple-choice science questions in the test set, respectively.
- OpenbookQA (Mihaylov et al., 2018): questions requiring multi-step reasoning, use of additional commonsense knowledge, and rich text comprehension. There are 500 questions in the test set.

Table 6: Description of Datasets used in experiments.

| Task Name | Domain | # Train | # Test | Task Type |
|---|---|---|---|---|
| BoolQ | Wikipedia | 9,427 | 3,270 | Text Classification |
| ARC-E | Natural Science | 2,250 | 2,380 | Question Answering |
| ARC-C | Natural Science | 1,120 | 1,170 | Question Answering |
| OpenBookQA | Science Facts | 4,957 | 500 | Question Answering |
| PIQA | Physical Interaction | 16,100 | 1,840 | Question Answering |
| SIQA | Social Interaction | 33,410 | 1,954 | Question Answering |
| HellaSwag | Video Caption | 39,905 | 10,042 | Sentence Completion |
| WinoGrande | Winograd Schemas | 9,248 | 1,267 | Fill in the Blank |

## A.2  VISUAL INSTRUCTION TUNING

Table 7 shows the details of the LLaVA training dataset. Table 8 shows the details of the test datasets.

Table 7: Detail of LLaVA-OneVision Dataset

| Datasets | Weight | Domain | Task Type |
|---|---|---|---|
| General | 36.1% | General | Various |
| Doc/Chart/Screen | 20.6% | Document | Question Answering, Chart Analysis |
| Math/Reasoning | 20.1% | Mathematics | Problem Solving, Reasoning |
| General OCR | 8.9% | OCR | Text Extraction, Recognition |
| Pure Language | 14.3% | Language | Text Generation, Language Modeling |

## A.3  DIFFUSION GENERATION

**Dataset Composition**  Each entry in the dataset consists of two keys: `image` and `text`. The `image` field contains a JPEG image loaded as a PIL object with variable dimensions, while the `text` field provides a descriptive caption corresponding to the image content. Only a `train` split is provided, indicating that the dataset is primarily intended for training purposes.

Table 8: Description of LLaVA test datasets

| Task Name | Domain | Task Type |
|-----------|--------|-----------|
| MMBench | Vision-Language | Fine-grained ability evaluation |
| MMVet | Multimodal | Integrated capability evaluation |
| MME | Multimodal | Comprehensive evaluation |
| ChartQA | Vision-Language | Question Answering about Charts with Visual and Logical Reasoning |
| AI2D | Vision-Language | Diagram Understanding and Question Answering |
| DocVQA | Vision-Language | Visual Question Answering on Document Images |
| MathVista | Vision-Language | Mathematical Reasoning in Visual Contexts |

**Caption Generation with BLIP**   To enrich the textual descriptions and improve the semantic alignment between images and captions, the original Pokémon images were processed through a pre-trained BLIP model. This model is capable of generating rich, context-aware captions that accurately describe the visual content. These generated captions serve as the textual conditioning input for training diffusion-based text-to-image models.

# B   EXPERIMENTAL SETUP

## B.1   COMMONSENE REASONING

Table 9 shows the detailed hyperparameters for commonsense reasoning tasks when fine-tuning the LLaMA3-8B.

Table 9: The hyperparameters for various methods on the commonsense reasoning tasks.

| Hyperparameter | LoRA | LoKr | AdaLoRA | HydraLoRA | ReLoRA | MoLA | LoRAMoE | MixLoRA |
|----------------|------|------|---------|-----------|--------|------|---------|---------|
| Rank $r$ | | | | 16 | | | | |
| $\alpha$ | | | | 32 | | | | |
| Dropout | | | | 0.05 | | | | |
| Target module | | | q, k, v, up, down | | | | q, k, v, o, gate, up, down | |
| #Experts | | - | | 4 | | | 8 | |
| Top-K | | - | | dense | | | 2 | |

## B.2   VISUAL INSTRUCTION TUNING

Table 10 shows the detailed hyperparameters for Visual Instruction Tuning when fine-tuning the LLaVA1.5-7B.

Table 10:  The hyperparameters for various methods on the Visual Instruction Tuning tasks.

| Hyperparameter | Single-LoRA | MoLE | HydraLoRA | ReLoRA |
|----------------|-------------|------|-----------|--------|
| Rank $r$ | | 32 | | |
| $\alpha$ | | 64 | | |
| Batch size | | 1 | | |
| Epochs | | 1 | | |
| Learning rate | | 2e-4 | | |
| Target module | | q, k, v, o, gate, up, down | | |
| #Experts | | 3 | | |

## B.3   DIFFUSION GENERATION

Table 11 shows the detailed hyperparameters for the Diffusion Generation task when fine-tuning the stable-diffusion v1.5. For GPT evaluation, refer to 12 and 13.

Table 11: The hyperparameters for various methods on the Diffusion Generation tasks.

| Hyperparameter | Single-LoRA | HydraLoRA | ReLoRA |
|:---:|:---:|:---:|:---:|
| Rank $r$ | | 4 | |
| $\alpha$ | | 8 | |
| Batch size | | 1 | |
| Steps | | 20000 | |
| Learning rate | | 1e-4 | |
| Target module | | q, k, v, o | |
| #Experts | | 3 | |

Table 12: Image Evaluation Criteria

| Criteria | Description |
|:---:|:---|
| Overall Quality | • Is the image clear and complete without obvious blur, noise or errors?
• Are the colors natural and harmonious, fitting the theme and scene? |
| Detail Richness | • Does the image have rich details in the subject and background?
• Are the details realistic and logically consistent with reality (if the theme is a real-life scene)? |
| Theme Consistency | • Does the image accurately reflect the given theme or description?
• Is there any deviation from the theme or unexpected content? |
| Creativity & Uniqueness | • Does the image show unique creativity or perspective?
• Are there novel elements or composition methods? |
| Style Matching | • Does the image match the specified style (such as realism, cartoon, oil painting, etc.)?
• Is it consistent with the target style? |
| Emotional Expression | • Can the image convey a certain emotion or atmosphere?
• Does it resonate with the audience? |
| Technical Performance | • Does the image demonstrate good generation technology, such as lighting and perspective?
• Are there any obvious generation errors or flaws? |

Table 13: Scoring Criteria

| Score | Description |
|:---|:---|
| 10 points | Perfect, almost flawless, exceeding expectations. |
| 8-9 points | Excellent, with a few minor flaws, but overall outstanding. |
| 6-7 points | Good, meeting expectations but with room for improvement. |
| 4-5 points | Average, with many problems that need improvement. |
| 2-3 points | Poor, not meeting expectations and requiring major adjustments. |
| 1 point | Very poor, almost unacceptable. |

## C    LIMITATION

Although the proposed `ReLoRA` achieves a good balance between parameter efficiency and model expressiveness, the current study focuses exclusively on parameter-efficient fine-tuning (PEFT) approaches, particularly those based on LoRA. While the method demonstrates strong performance in fine-tuning tasks, its effectiveness has not been evaluated on other efficient adaptation paradigms such as prompt-tuning, prefix-tuning, or fully frozen training strategies. Additionally, the framework has only been applied in the downstream fine-tuning phase; its potential applicability during the pre-training stage remains an open question for future exploration. Future work may explore more efficient routing mechanisms, hybrid PEFT frameworks, and extensions to the pre-training phase to further improve both efficiency and generalization.

## D    THE USE OF LARGE LANGUAGE MODELS

We used LLMs solely as a writing-assistance tool to polish our paper (grammar, wording, concision, and minor LaTeX formatting). The LLM did not contribute to research ideation, problem formulation, method design, experiments, data analysis, results, or conclusions, and it was not used to generate citations or technical content. All suggestions were reviewed and, when adopted, edited by the authors, who take full responsibility for the paper's content; no proprietary data beyond the manuscript text was shared with the tool.

