# OpenReview forum: "Less is More: Resource-Efficient Low-Rank Adaptation"
_ICLR.cc/2026/Conference — ICLR 2026 Conference Withdrawn Submission_

### Official Review · Reviewer_Fhov · 2025-10-26

**Soundness:** 3
**Presentation:** 2
**Contribution:** 2
**Rating:** 2
**Confidence:** 4

**Summary:**

This paper introduces ReLORA, a lightweight framework designed to overcome the parameter redundancy and task interference limitations inherent in standard LoRA. ReLORA's architecture is built on two core innovations: a unified asymmetric architecture that utilizes a single, globally shared A matrix across all transformer layers to encode common knowledge, which is complemented by specialized, layer-specific B matrices to capture fine-grained adaptations. Additionally, the framework features a dynamic Reducer mechanism that, during training, intelligently freezes less important B matrices based on calculated importance scores, allowing for a flexible trade-off between computational resources and model performance. The paper validates this approach across diverse modalities—including language, vision-language, and diffusion models—demonstrating that ReLORA consistently outperforms standard LoRA and other strong baselines on tasks like commonsense reasoning, visual instruction tuning, and image generation, all while achieving superior parameter efficiency.

**Strengths:**

- Superior Parameter Efficiency via Asymmetric Architecture: The paper introduces a unified asymmetric architecture that effectively tackles parameter redundancy. By employing a single, globally shared A matrix across all layers, it captures common, generalizable knowledge , while using specialized, layer-specific B matrices to learn fine-grained task details. This design drastically reduces the total number of trainable parameters compared to standard LoRA and its variants, achieving SOTA results with significantly fewer parameters.

- Dynamic Resource-Aware Training: ReLoRA introduces a Reducer mechanism, which is a dynamic training strategy, not a post-training pruning method. This component intelligently calculates the importance of different B matrices during training and selectively freezes the least contributive ones . This allows users to flexibly trade off the computational budget and model performance, enabling efficient adaptation even under stringent resource constraints.

- Effective Mitigation of Task Interference: The framework is explicitly designed to handle complex, heterogeneous datasets where standard LoRA suffers from task conflicts . By decoupling general knowledge (in the shared A matrix) from specialized knowledge (in the B-heads), ReLoRA effectively reduces conflicting objectives. This is strongly validated in multimodal experiments, where ReLoRA avoids the significant performance collapse that vanilla LoRA experiences when trained on diverse data mixtures.

- Broad Generalizability Across Modalities: The paper demonstrates that ReLoRA's benefits are not limited to language models. The method is successfully applied and shown to consistently outperform strong baselines across three distinct and diverse modalities: commonsense reasoning in LLMs, visual instruction tuning in multimodal models, and high-fidelity image generation with diffusion models.

**Weaknesses:**

- `Limited Innovation`: The ReLoRA method is overly simplistic, and its core motivation ($i.e.$, low-resource LoRA research [1-3]) has already been extensively explored.
- `Limited Presentation`: Figure 4 fails to adequately explain the proposed ReLoRA. It is highly confusing whether the number of B matrices is singular, multiple per layer, or determined by the single-task vs. multi-task experimental setup. Furthermore, the mechanism of the importance score vector is unclear; how importance is judged and the utility of the resulting sampling distribution are not elaborated upon.
- `Limited Comparison`: ReLoRA's approach of decoupling the LoRA A and B matrices across different layers restricts the applicability of other, more efficient LoRA initialization methods [4-6].
- `Limited Code Availability`: The code in the supplementary material is disorganized. The provided scripts are non-functional due to numerous missing dependencies and files, many of which seem to be included merely as placeholders. This severely impedes further review and reproducibility.

[1] LoRA-XS: Low-Rank Adaptation with Extremely Small Number of Parameters

[2] VeRA: Vector-based Random Matrix Adaptation

[3] LoRA-FA: Memory-efficient Low-rank Adaptation for Large Language Models Fine-tuning

[4] PiSSA: Principal Singular Values and Singular Vectors Adaptation of Large Language Models

[5] CoLA: Collaborative Low-Rank Adaptation

[6] LoRA-GA: Low-Rank Adaptation with Gradient Approximation

**Questions:**

See Weaknesses.

---

### Official Review · Reviewer_t7Ep · 2025-10-29

**Soundness:** 3
**Presentation:** 3
**Contribution:** 2
**Rating:** 4
**Confidence:** 5

**Summary:**

This paper proposes E-LoRA, a variation of LoRA for PEFT.

Its general idea is to sharing one matrix across layers and pruning only a subset of adapters per step with a small router.

The proposed method shows clear improvement on the modalities and tasks across language, VLM, and diffusion.

**Strengths:**

+ The general idea to *general idea is to sharing one matrix across layers and pruning only a subset of adapters per step with a small router.* is pratical and straight-foward.

+ Clear improvement over the baselines.

+ Overall this paper is easy to follow.

+ This paper covers the improvement across multi-modalities.

**Weaknesses:**

- The technique novelties of the proposed method remains limited. Specifically, prior work (e.g., AdaLoRA, DyLoRA) has already shared or frozen matrix to remove inter-matrix redundancy and shown dynamic or adaptive budgets across layers.

- The compared state-of-the-art PEFT methods are significantly missing. Some more recent and much stronger PEFT methods are mssing for comparison, for example:

[1] DoRA: Weight-Decomposed Low-Rank Adaptation. ICML 2024.

[2] VeRA: Vector-based Random Matrix Adaptation. ICLR 2024.

[3] Foura: Fourier low-rank adaptation. NeurIPS 2024.

[4] SSH: Sparse Spectrum Adaptation via Discrete Hartley Transformation. NAACL 2024.

- The Router in the proposed method is described as “lightweight”, but its gating granularity (token, head, or sequence), regularization and throughout are not quantified.

- No activation memory or KV-cache impact is reported. Please note, for sequence models this dominates inference costs.

- A more breakdown analysis on the effciency should be reported, for example, on the FLOPs of the adapter, multi-head and etc.

- The ablation study is not extensive. Ideally, the impact of each component should be considered as a setting and report the outcomes.

Meanwhile, there is also some minor issue to be fixed:

- The impact of the temperature parameter is not analyzed.

- Some figures mix parameter count.

- This paper only discusses training-time only. The inference-time and other inference behaviour should be considered.

**Questions:**

Please refer to the weakness section and address these point-by-point.

---

### Official Review · Reviewer_R4AV · 2025-10-30

**Soundness:** 3
**Presentation:** 3
**Contribution:** 3
**Rating:** 6
**Confidence:** 3

**Summary:**

This paper introduces ReLoRA (Resource-Efficient Low-Rank Adaptation), a novel parameter-efficient fine-tuning framework that revisits LoRA through the lens of cross-layer and inter-matrix redundancy. The authors identify that conventional LoRA suffers from duplicated low-rank subspaces and interference across heterogeneous domains. ReLoRA addresses these issues via two key designs:
* A Unified Asymmetric Architecture, where all Transformer layers share a single global down-projection matrix $A$, while layer-specific up-projection matrices $B_i$ are dynamically selected by a router network;
* A Reducer module, which dynamically freezes less important $B_i$ matrices based on layer contribution scores, thereby balancing performance and resource usage during training.
Experiments across language reasoning (LLaMA3-8B), multimodal instruction tuning (LLaVA-7B), and diffusion image generation demonstrate that ReLoRA outperforms LoRA and strong variants like HydraLoRA and GraphMoE, achieving comparable or better accuracy with 40–60% fewer trainable parameters.

**Strengths:**

* *Clear motivation and solid empirical grounding*: The paper convincingly identifies inter-matrix and intra-layer redundancies in LoRA through well-designed empirical observations.
* *Simple yet generalizable design*: The unified $A$+dynamic $B$ mechanism is conceptually elegant and applicable to various architectures (LLMs, MLLMs, diffusion models) without invasive changes.
Strong empirical validation – Results are consistent across diverse modalities with detailed ablations (e.g., drop-ratio, number of $B$ heads, importance-based vs. random pruning).
* *Improved efficiency–performance trade-off*: Demonstrates that ReLoRA attains HydraLoRA-level or better performance with about half the tunable parameters, confirming its efficiency claim.
* *Readable and well-structured*: The paper’s presentation is coherent; figures and tables are clear, and the motivation-to-method transition is smooth.

**Weaknesses:**

* *Incomplete resource-efficiency evaluation*: The “resource-efficient” claim mainly relies on parameter count and training time; other dimensions such as GPU memory footprint, FLOPs under varying K, or energy consumption are not analyzed. What's more, only three model is adopted.
* *Generalization explanation*: The paper attributes performance gains to shared-A capturing “global knowledge,” but lacks deeper analysis (e.g., probing studies or representation overlap) to validate this hypothesis.

**Questions:**

* How sensitive is the performance to the choice of $K$ (the number of active layers) and the frequency of Reducer updates? Is there any adaptive rule to select $K$ automatically?
* Have the authors measured runtime or memory savings during inference (not just training)?
* Does sharing a global $A$ harm specialization in deeper layers for tasks requiring hierarchical representations (e.g., syntax vs. semantics)?
* Could ReLoRA be combined with rank-adaptive or quantized LoRA variants?
* For multimodal tuning, does the same shared $A$ span text and visual projections, or are separate $A$ matrices used per modality?
* How will the performance be on other models?

---

### Official Review · Reviewer_N1Ed · 2025-11-04

**Soundness:** 3
**Presentation:** 3
**Contribution:** 3
**Rating:** 4
**Confidence:** 4

**Summary:**

This paper proposes a parameter-efficient finetuning method for transformers. It uses a single downprojection matrix A across layers and then prunes B matrices for use in different layers to balance expressivity and parameter count. Experiments are thorough and results are strong with some details missing.

**Strengths:**

* good comparison against existing works
* compared using current architectures (like llama-3)
* proposed method is very parameter-efficient.
* experiments cover several domains from LLMs to VLMs to diffusion models
* includes training complexity/cost analysis which is favorable

**Weaknesses:**

* the methodname "ReLora" is already taken from a fairly well-cited work of 2023. published at iclr
* Limited novelty. sharing weights across layers is not novel (e.g. HydraLoRA), using a MoE like router has also been used in LoRAMoE and other works. What remains as novelty seems to be the "reducer" part of the method which only aids training and does not have any effect on downstream use.
* unclear hyperparameter selection. LoRA and all the PEFT methods are extremely dependend on correct and fair hyperparameter choice. The paper does not mention what hparam grid was used across the methods and how the final selection took place (some methods benefit from higher learning rate etc).
* Family of LoRA methods missing: the works that use random matrices are extremely parameter-efficient and not compared against nor mentioned in related works. these include BOFT/NOLA, VeRA and LoReTTA, for example

**Questions:**

* hyperaparameter-grid selection and details on final model reporting (see above)
* performance against random-matrix based approaches
* details of difference compared to existing works such as MoELoRA should go into related works.
* For the VLM experiment: a full-finetune baseline is missing.

---

### Note · Authors · 2025-11-29

I have read and agree with the venue's withdrawal policy on behalf of myself and my co-authors.